# Molecular Characterization, Recombinant Expression, and Functional Analysis of Carboxypeptidase B in *Litopenaeus vannamei*

**DOI:** 10.3390/genes16010069

**Published:** 2025-01-09

**Authors:** Hongmei Li, Hai Lin, Hao Yang, Chunhua Ren, Yi He, Xiao Jiang, Ting Chen, Chaoqun Hu

**Affiliations:** 1School of Engineering, Guangzhou College of Technology and Business, Guangzhou 510850, China; hongmeili22@126.com (H.L.); hailinaug@126.com (H.L.);; 2CAS Key Laboratory of Tropical Marine Bio-Resources and Ecology, South China Sea Institute of Oceanology, Chinese Academy of Sciences, Guangzhou 510301, China; 3Key Laboratory of Applied Marine Biology of Guangdong Province and Chinese Academy of Sciences (LAMB), South China Sea Institute of Oceanology, Chinese Academy of Sciences, Guangzhou 510301, China; 4South China Sea Bio-Resource Exploitation and Utilization Collaborative Innovation Center, Guangzhou 510301, China

**Keywords:** Pacific white shrimp, molecular characterization, enzymatic activity, functional analysis

## Abstract

**Background/Objectives:** The Pacific white shrimp (*L. vannamei*) is economically significant, and its growth is regulated by multiple factors. Carboxypeptidase B (CPB) is related to protein digestion, but its gene sequence and features in *L. vannamei* are not fully understood. This study aimed to explore the molecular and functional properties of CPB in *L. vannamei*. **Methods:** The Lv-CPB gene was cloned, and bioinformatics analysis, qRT-PCR, in situ hybridization, recombinant protein expression in *Escherichia coli*, and an enzyme activity assay were performed. **Results:** The Lv-CPB gene is 1414 bp long with a 1263 bp ORF encoding a 420-amino-acid protein. It is stable, hydrophilic, and is highly expressed in the hepatopancreas. The recombinant protein was efficiently expressed with a molecular weight of about 47 kDa. The optimal pH and temperature for Lv-CPB were 8.0 and 50 °C, respectively. **Conclusions:** This study revealed the molecular and functional characteristics of Lv-CPB, providing insights into its role in shrimp digestion, as well as suggestions for improving aquaculture practices.

## 1. Introduction

Carboxypeptidase B (CPB) is a metallopeptidase belonging to the M14 family of exopeptidases, which are primarily responsible for cleaving essential amino acids, such as arginine and lysine, from the C-terminus of peptides and proteins [1]. This enzyme plays a pivotal role in various biological processes, including protein degradation [2], the maturation of bioactive peptides [3], and the regulation of peptide hormones [4]. CPB is involved in the final stages of proteolytic processing, complementing the action of other proteases by removing terminal residues from substrates, thus contributing to the regulation of diverse physiological functions [5]. In vertebrates, CPB was first characterized in the mammalian pancreas, where it contributes to digestive processes, hydrolyzing the peptides produced by trypsin and chymotrypsin [6]. In addition to its digestive functions, CPB is implicated in pathways such as coagulation, inflammation, and wound healing, making it a versatile enzyme with broad biological significance [7]. CPB enzymes are evolutionarily conserved across various organisms, including bacteria, fungi, plants, and animals, suggesting their fundamental role in cellular and metabolic activities [8,9,10]. At present, extensive research has been conducted on CPB in vertebrates, especially in mammals. Additionally, some studies have detected carboxypeptidase activity in the digestive systems of invertebrates, such as *Aedes aegypti* [11,12,13,14]. However, a further understanding is still needed regarding the presence and function of CPB in invertebrates, especially in economically important species like crustaceans. Taking Procambarus clarkii as an example, currently, only sequence analysis has been reported, and the comprehensive molecular characterization of its CPB is relatively lacking.

*L. vannamei* is one of the most widely cultured crustaceans, contributing significantly to global aquaculture production [15]. The digestive efficiency of *L. vannamei* is a critical factor in its growth and development, making the study of its digestive enzymes crucial for optimizing shrimp farming practices [16,17]. Research on digestive enzymes in *L. vannamei* has predominantly focused on carbohydrates and lipases; however, proteolytic enzymes, particularly CPB, which are essential for protein digestion and nutrient absorption, still need to be explored. Understanding CPB’s molecular characteristics and functional roles in *L. vannamei* could provide valuable insights into its digestive physiology and contribute to developing more efficient feed formulations, enhancing growth rates and feed conversion efficiency in aquaculture.

In this study, we investigated carboxypeptidase B’s molecular and functional characteristics in *L. vannamei* (Lv-CPB). The Lv-CPB gene was cloned and sequenced, followed by bioinformatic analysis to predict its structural features and evolutionary relationships. We assessed its expression across various tissues and developmental stages, with specific localization in the hepatopancreas determined through in situ hybridization. Additionally, the recombinant Lv-CPB protein was expressed in a heterologous *E. coli* system, enhanced by codon optimization, and its enzymatic activity was evaluated under different temperature and pH conditions. This study seeks to enhance our understanding of proteolytic enzymes in crustaceans and offers crucial data to support the use of recombinant protein production systems for investigating marine-sourced enzymes. These results are of practical importance for improving nutrient utilization efficiency in shrimp aquaculture and advancing sustainable production practices.

## 2. Materials and Methods

### 2.1. Animals

For molecular cloning and tissue distribution studies, healthy *L. vannamei* samples weighing 45.0 ± 5.0 g were collected from the Jinyang aquaculture base in Maoming, Guangdong Province, China. The shrimp were placed in 100 L tanks with a water salinity of 30‰, a pH of 8.0, and a water temperature of 28 °C. They were fed three times daily and acclimated for one week under continuous aeration.

### 2.2. Cloning and Sequencing of the L. vannamei CPB Gene

Total RNA extraction and reverse transcription from *L. vannamei* followed previously established protocols [18]. Based on the *L. vannamei* transcriptome data and the predicted *L. vannamei* CPB sequence from NCBI (XP_027208360.1), primers were designed using Primer Premier 7.0 (Appendix A) for the PCR amplification of the Lv-CPB ORF sequence. The PCR reaction system (50 µL) contained 0.5 µL PrimeSTAR, 10 µL 5× PS Buffer, 4 µL dNTP, 1 µL each of the Lv-CPB F/R primers (10 µmol/L), 1.5 µL cDNA, and 32 µL ddH_2_O. The amplification conditions were as follows: pre-denaturation at 94 °C for 5 min, followed by 35 cycles of denaturation at 94 °C for 30 s, annealing at 55 °C for 45 s, and extension at 72 °C for 1 min, with a final extension at 72 °C for 10 min. The target PCR fragments were recovered, ligated into pET28a vectors, and then transformed into *E. coli*. Positive clones were selected and sent to Sangon Biotech (Shanghai, China) Co., Ltd. for sequencing analysis. The sequencing results were compared to database sequences to verify their accuracy. Sequences of the Lv-CPB gene and protein are presented in Appendix A.

### 2.3. Bioinformatics Analysis of the Lv-CPB Gene

The Lv-CPB gene’s full ORF was predicted using the online ORF finder tool (http://www.bioinformatics.org/sms2/orf_find.html, accessed on 10 October 2024). The physicochemical properties of the Lv-CPB protein were analyzed using the ExPASy-ProtParam tool-4.0 (https://web.expasy.org/protparam/, accessed on 10 October 2024), and functional site analysis was performed using SoftBerryPsite-2.0 (http://www.softberry.com/, accessed on 10 October 2024). The structural domains of Lv-CPB were predicted with the ScanProsite program-4.0 (http://prosite.expasy.org/, accessed on 10 October 2024). Signal peptide prediction was conducted with the SignalP 5.0 Server (https://services.healthtech.dtu.dk/service.php?SignalP-5.0, accessed on 10 October 2024), while transmembrane structures were predicted using TMHMM-2.0 (https://services.healthtech.dtu.dk/services/TMHMM-2.0/, accessed on 10 October 2024) and Euk-mPLoc 2.0 (http://www.csbio.sjtu.edu.cn/bioinf/euk-multi-2/, accessed on 10 October 2024). A 3D model of Lv-CPB was constructed using Swiss-model software from the SWISS-MODEL server-14.0 (https://swissmodel.expasy.org/, accessed on 10 October 2024) and visualized with the Schrodinger Discovery Suite 2022-04. Multiple sequence alignments of CPB across species were generated using DNAMAN 6.0 software, and a phylogenetic tree was constructed using the neighbor-joining method in MEGA 7.0 software [19].

### 2.4. Tissue Distribution of Lv-CPB mRNA

The tissue distribution of Lv-CPB mRNA was quantitatively analyzed across three individual shrimp. The selected tissues included the brain, eyestalk, gills, hemolymph, hepatopancreas, heart, abdominal ganglion, thoracic ganglion, testis, ovary, stomach, intestine, and muscle. Shrimp tissues were processed using 1 mL of TRIzol reagent (Invitrogen, Schanghai, China). After this, total RNA was extracted and reverse transcribed into cDNA using the PrimeScript RT reagent Kit (Takara). Specific primers—q-CPB-F and q-CPB-R—were designed based on the Lv-CPB mRNA sequence (Appendix A). Quantitative PCR was conducted using the TaKaRa SYBR Premix Ex Taq Kit, with the reaction mixture containing 10.0 µL SYBR Premix Ex Taq, 0.4 µL of each primer, 2 µL cDNA, and 7.2 µL ddH_2_O. RT-PCR was performed on the Thermal Cycler Dice Real Time System III (TaKaRa, Daliang, Chia) with a total reaction volume of 20 µL. Amplification was carried out using a two-step method consisting of 40 cycles at 95 °C for 5 s and 60 °C for 30 s. As described in our previous research, Lv β-actin was used as the internal reference gene [12]. The expression levels of Lv-CPB in each tissue were calculated using the 2^−ΔΔCt^ method.

### 2.5. In Situ Hybridization of Lv-CPB in the Hepatopancreas

For the in situ hybridization (ISH) of Lv-CPB in the hepatopancreas, tissue samples were prepared following previously described protocols [18]. A DIG-labeled DNA probe targeting Lv-CPB was synthesized using the PCR DIG Probe Synthesis Kit (Sigma-Aldrich, Germany) with primers p-CPB-F and p-CPB-R (Appendix A). Shrimp hepatopancreas samples were collected and processed for ISH. The diaminobenzidine (DAB) method detected the ISH signal through incubation with anti-DIG antibodies conjugated to horseradish peroxidase (HRP), and the nuclei were counterstained with hematoxylin. Image observation was carried out using the Case Viewer system. Negative controls, in which the Lv-CPB DNA probe was omitted, were included, and parallel histological staining with hematoxylin and eosin (H&E) was performed for tissue structure examination [19].

### 2.6. Expression and Purification of Recombinant Lv-CPB

The codons for the first 20 amino acids following the signal peptide in the Lv-CPB ORF were optimized without altering the original amino acid sequence. PCR amplification was performed using the primers listed in Appendix A, and the product was subsequently subcloned into the pET28a vector, which had been linearized with *ScaI* and *XhoI* using the ClonExpress II One Step Cloning Kit (Vazyme, Nanjing, China). The transformation, cultivation, and induction of Lv-CPB expression in the pET28a recombinant expression vector and protein purification were conducted as previously described [20]. Protein concentration was determined using the Lowry method, with bovine serum albumin as the standard [21].

### 2.7. Activity Assay of Lv-CPB

Carboxypeptidase activity was measured following the method described by Folk et al. [22] using hippuryl-L-arginine as the substrate and 0.025 M Tris-HCl buffer containing 0.1 M NaCl at pH 7.65. One unit of enzyme activity (U) was defined as the amount of enzyme required to hydrolyze one µmol of substrate per minute at 25 °C, with specific activity expressed in U/mg. After adding 100 µL of enzyme solution to the 3 mL reaction system containing a substrate concentration of 1 μmol/L, the mixture was immediately shaken, the baseline was adjusted, and the change in absorbance at 254 nm was measured over 5 min.

Recombinant Lv-CPB was incubated with 1 mmol/L hippuryl-L-arginine at 25 °C in buffers of varying pH (3.0, 4.0, 5.0, 6.0, 6.5, 7.0, 7.5, 8.0, 8.5, 9.0, 10.0, 11.0, and 12.0) for 5 min, and the relative activity was measured to determine the optimal reaction pH. The enzyme’s relative activity at different temperatures (10 °C, 20 °C, 25 °C, 30 °C, 35 °C, 40 °C, 45 °C, 50 °C, 55 °C, 60 °C, 65 °C, 70 °C, 75 °C, and 80 °C) was assessed at pH 7.65, with a reaction time of 5 min, to determine the optimal reaction temperature.

## 3. Results

### 3.1. Molecular Cloning and Structural Characterization of Lv-CPB

Based on transcriptome prediction [18], cloning was completed through PCR and verified by sequencing analysis, and the ORF sequence of Lv-CPB was obtained (Figure 1A). The Lv-CPB gene has a length of 1414 bp, with an ORF of 1263 bp encoding 420 amino acid (aa) residues. The molecular weight of the Lv-CPB protein is 46.9 kDa, with a theoretical isoelectric point (pI) of 4.83 and an instability index of 28.33, classifying it as a stable, hydrophilic protein (grand average of hydropathy (GRAVY): −0.345) according to ExPASy-ProtParam analysis. The Lv-CPB protein contains a signal peptide, an activation peptide domain, and a functional enzymatic domain with no transmembrane regions. The signal peptide cleavage site is predicted between the 16th and 17th amino acids, specifically between alanine (ALA) and arginine–proline (RP), with a cleavage probability of 0.8255, indicating high prediction confidence.

Following the 3D modeling of the Lv-CPB protein, functional domain analysis revealed that the activation peptide domain (APD, 30-100 aa) forms a globular structure, followed by an extended α-helix, which shields the catalytic site from interacting with substrates. This suggests that the Lv-CPB protein is synthesized in an inactive precursor form and requires the proteolytic cleavage of the propeptide for enzyme activation [23,24]. The peptidase M14 carboxypeptidase domain (PMCD, 122-413 aa) is a zinc-dependent carboxypeptidase with a recognition site for the free C-terminal carboxyl group. It can hydrolyze a single amino acid from the C-terminus of polypeptide chains [25]. This domain contains two zinc-binding regions (170-192 aa and 304-314 aa) and a catalytic active site, where the zinc ion is essential for enzymatic activity [24] (Figure 1B,C).

### 3.2. Phylogenetic, Homology, and Structural Analysis

A multiple sequence alignment of Lv-CPB with CPB from other closely related species revealed that the aa sequence of Lv-CPB shares 34.08–53.75% identity with other known carboxypeptidases (Figure 2A). Despite the variation in sequence identity, the catalytic function, Zn^2+^ binding sites, and catalytic residues are highly conserved across carboxypeptidases from different animal species. The phylogenetic analysis based on the primary structure of Lv-CPB (Figure 2B) demonstrates the closest evolutionary relationship with *Procambarus clarkii* (crayfish) and *Homarus americanus* (lobster), showing evolutionary distances of 0.19822910 and 0.29667740, respectively, both with a bootstrap value of 0.9790, indicating the relative conservation of Lv-CPB among closely related species.

Furthermore, a comparison of the spatial structures of CPB proteins from *L. vannamei*, the related shrimp species *P. clarkii* and *H. americanus*, and the closely related crab species *Euphausia superba* revealed that the secondary structure and domain composition of CPB proteins are conserved in crustacean species. This suggests a similar functional mechanism. Additionally, all CPB proteins possess the activation peptide domain (APD), indicating that the regulatory mechanisms governing CPB function may also be conserved across these organisms (Figure 2C).

### 3.3. Expression Profiles of Lv-CPB in Different Tissues

The expression levels of the Lv-CPB gene in 12 different tissues of *L. vannamei*, including the brain, eyestalk, gills, hemolymph, hepatopancreas, heart, intestine, muscle, ovary, stomach, abdominal ganglion, testis, and thoracic ganglion, were analyzed using qRT-PCR. As shown in Figure 3A, CPB mRNA expression was detected in most of the selected tissues, with varying signal distribution, indicating the tissue-specific expression of the Lv-CPB gene. The highest expression level was observed in the hepatopancreas, followed by the hemolymph and brain, while expression in the ovary, thoracic ganglion, and abdominal ganglion was nearly undetectable. ISH was performed to confirm Lv-CPB localization within the hepatopancreas. The probe staining revealed that Lv-CPB is predominantly expressed in parenchymal cells, with lower expression levels in ductal cells. Moreover, Lv-CPB expression was higher in hepatopancreatic cells compared to the basement membrane, suggesting that Lv-CPB mainly originates from hepatopancreatic cells (Figure 3B).

### 3.4. Recombinant Expression and Optimization of Lv-CPB Protein

Lv-CPB protein was recombinantly expressed in *E. coli* in this phase. Due to the presence of numerous rare codons, particularly at the 5′ end of the Lv-CPB DNA sequence, protein expression efficiency was significantly reduced. To address this issue, codon optimization was performed, as shown in Figure 4A. Following optimization, the Lv-CPB gene was efficiently expressed in *E. coli*, with high concentrations of Lv-CPB protein detected in both the soluble supernatant (SU) and inclusion fraction (IF) of *E. coli* lysates (Figure 4B). SDS-PAGE analysis revealed that the recombinant protein had a molecular weight of approximately 47 kDa, consistent with the previous prediction. Subsequently, the effects of different isopropyl β-D-thiogalactopyranoside (IPTG) concentrations on protein expression were compared. The results demonstrated that high IPTG concentrations (1.0 mM) led to an 18.5% increase in expression levels compared to low IPTG concentrations (0.2 mM) based on grayscale analysis (Figure 4C). After the lysis of the expression strain, the Lv-CPB protein was purified using immobilized metal affinity chromatography (IMAC) and desalted through a PD-10 column, yielding a final high-purity protein product at a concentration of 3.27 mg/mL (Figure 4D) for subsequent studies.

### 3.5. Enzymatic Properties of Lv-CPB

*L. vannamei* is a species that is highly susceptible to fluctuations in environmental pH and temperature [26,27,28], which significantly affects the efficiency of commercial aquaculture. We conducted comparative analyses under various pH and temperature conditions to characterize the impact of pH and temperature on Lv-CPB’s enzymatic activity. Initially, Lv-CPB activity was measured by assessing the hydrolysis of hippuryl-L-arginine, following the assay kit instructions. The enzyme activity was defined as the amount of enzyme required to hydrolyze 1 μMol of substrate per minute at pH 7.65 and 25 °C, and specific activity was expressed in units per milligram (U/mg). The standard curve equation for Lv-CPB activity was determined as y = 0.3569 x + 0.049 (R^2^ = 0.9924) (Figure 5A).

Then, a pH gradient from 3.0 to 12.0 was established to assess the relative enzymatic activity of Lv-CPB under different pH conditions. The results showed that Lv-CPB exhibited the highest catalytic activity at pH 8.0, with 80% of its activity occurring within the pH range of 7.43 to 8.25, indicating optimal activity under slightly alkaline conditions (Figure 5B). Subsequently, a temperature gradient ranging from 10 °C to 80 °C was applied to evaluate Lv-CPB activity at different temperatures. The enzyme displayed maximal catalytic activity at 50 °C, with 80% of its activity occurring within the temperature range of 46.03 °C to 52.75 °C, followed by a sharp decline in activity at temperatures exceeding 55 °C (Figure 5C). These observations suggest that Lv-CPB exhibits a degree of heat tolerance. In summary, we found that Lv-CPB activity is more significantly influenced by pH than temperature, suggesting that Lv-CPB may contribute to *L. vannamei*’s sensitivity to environmental pH changes.

## 4. Discussion

In this study, we cloned and characterized the Lv-CPB gene from *L. vannamei*, providing valuable insights into its molecular structure and enzymatic function. Sequence analysis revealed that Lv-CPB encodes a 420 aa protein with a molecular weight of 46.9 kDa, which agrees with the predicted values. Genomic DNA (gDNA) was extracted, and primers were designed based on the exon sequences to amplify and sequence the full-length gene. The results revealed that the Lv-CPB gene spans 12,282 bp in the gDNA, comprising nine exons and eight introns (Appendix A). The protein features a signal peptide and key functional domains, including an APD and the PMCD, both essential for its enzymatic activity. The highly conserved zinc-binding sites within the PMCD, crucial for catalysis, are consistent with the general characteristics of carboxypeptidases across various species. Phylogenetic analysis further confirmed that Lv-CPB shares a close evolutionary relationship with *P. clarkii* and *H. americanus*, indicating a conserved functional mechanism among these species [29,30].

Phylogenetic and homology analyses confirmed the evolutionary conservation of CPB proteins in crustaceans despite sequence identities between Lv-CPB and other species ranging from 34.08% to 53.75%. The catalytic residues, Zn^2+^ binding sites, and enzymatic domains remained highly conserved, indicating that CPB proteins retain similar structural and functional features across species. Further analysis of secondary structure and domain composition supports the idea that CPB proteins in crustaceans share common functional mechanisms, suggesting they operate in comparable physiological contexts among related species. When comparing the roles of CPB in mammals versus shrimp, several similarities and differences emerge. In both groups, CPB is involved in protein digestion, primarily by hydrolyzing basic amino acids from the C-terminus of peptides. However, while the digestive role of CPB in mammals has been well documented in the pancreas, where it aids in protein digestion and other metabolic processes, the role of CPB in shrimp, particularly in *L. vannamei*, is less understood but similarly critical for digestion in the hepatopancreas. The similarities and differences in CPB functions between mammals and shrimp are summarized in Appendix A. This comparison underscores the evolutionary conservation of CPB’s core catalytic functions across taxa while highlighting adaptations to specific physiological and environmental contexts [31,32,33].

qRT-PCR analysis revealed that Lv-CPB expression is tissue-specific, with the highest levels in the hepatopancreas, followed by the hemolymph and brain, while reproductive tissues showed minimal expression. This highlights Lv-CPB’s key role in digestion, further supported by ISH, which localized its expression into hepatopancreatic parenchymal cells. However, the CPB gene has been identified to significantly impact reproduction in certain organisms. For example, in *Ophraella communal*, CPB shows a strong male-biased expression; its expression is upregulated in the bursa copulatrix of mated females, and the knockdown of the CPB gene in males leads to a decrease in insect fecundity [34].

Carboxypeptidase is a kind of exopeptidase that can specifically release free aa one by one, starting from the C-terminus of the peptide chain. It is widely used in fields such as medicine and food [35]. Commercial carboxypeptidase is acquired mainly from the pancreas of pigs and cows [36,37]. Recombinant Lv-CPB expression in *E. coli* was optimized through codon modification, leading to significantly improved yields, as confirmed by SDS-PAGE. To further investigate the effect of IPTG concentration on protein production, two concentrations were compared—a low concentration of 0.2 mM and a high concentration of 1.0 mM. The higher IPTG concentration increased protein production by 18.5%. The results of the enzyme activity assay demonstrated that CPB from *L. vannamei* exhibited optimal activity at pH 8.0. Previous studies have indicated that metal carboxypeptidases possess significant activity under neutral or slightly alkaline conditions, further confirming that Lv-CPB possesses the typical characteristics of metalloproteases [38]. The enzyme activity assay results also revealed that the optimal reaction temperature for this enzyme was 50 °C, comparable to the optimal temperatures of carboxypeptidases purified from porcine pancreas and bovine pancreas [39,40]. This optimal temperature was higher than that of cold-adapted enzymes such as those from *E. superba*, whose optimal temperature was 30 °C and still maintained a certain catalytic activity within the temperature range of 0–10 °C. This indicates that Lv-CPB has a certain degree of heat tolerance [41]. In the research field of marine invertebrates, the exploration of CPB has been continuously deepening. Previous studies have found that in marine invertebrates, the starfish (*Asterias amurensis*) was the first species on which a comprehensive characterization and protein function analysis of CPB was conducted. CPB was purified from its pyloric ceca. This enzyme was nearly homogeneous in polyacrylamide gel electrophoresis, and its molecular weight was approximately 34,000. The optimum pH for its hydrolysis of benzoyl-glycyl-L-arginine was approximately 7.5, and the optimum temperature was approximately 55 °C, which are different from the molecular weight, optimum pH, and temperature conditions of Lv-CPB in this paper, reflecting the diversity of the functional characteristics of carboxypeptidase B among different marine invertebrate species [42].

Although this study provides important insights into the molecular and functional characteristics of Lv-CPB in *L. vannamei*, some areas could benefit from further exploration. The in vivo functionality of Lv-CPB and its role under different physiological conditions still need to be fully addressed, and additional research may help to clarify its broader biological significance. While the recombinant expression in *E. coli* was optimized, future studies could explore Lv-CPB’s behavior in more natural or in vivo systems. These areas for improvement could guide the focus of future research.

This study provides the first comprehensive molecular and functional analysis of Lv-CPB in *L. vannamei*, revealing its tissue-specific expression and sensitivity to environmental factors. The recombinant expression and characterization of Lv-CPB lay the groundwork for future research on its role in enhancing nutrient absorption and aquaculture productivity. Furthermore, insights into its enzymatic properties have practical implications for optimizing shrimp farming practices, particularly in managing environmental pH and temperature to improve growth and health in *L. vannamei*.

## Figures and Tables

**Figure 1 genes-16-00069-f001:**
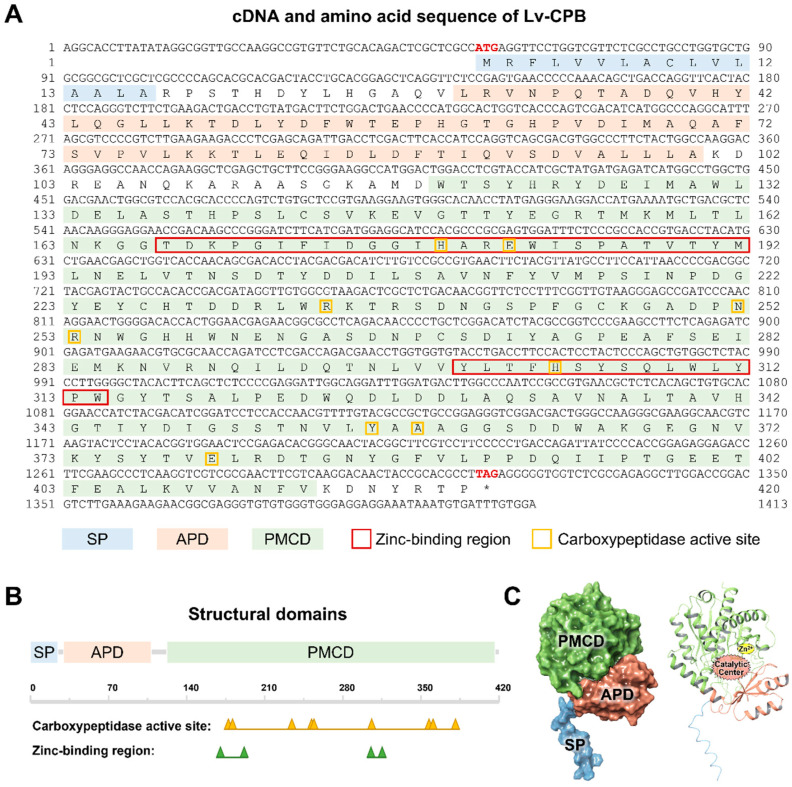
Full-length cDNA sequence and structural analysis of Lv-CPB. (**A**) The complete cDNA sequence of Lv-CPB and its corresponding translated amino acid sequence. The start codon ATG and stop codon TAG are highlighted in red. The signal peptide (SP) is shaded in blue, the activation peptide domain (APD) is marked in red, and the peptidase M14 carboxypeptidase domain (PMCD) is marked in green. A red box indicates the zinc ion-binding region, while the catalytic active site residues are enclosed in a yellow box. (**B**) Schematic representation of the structural domains of the Lv-CPB protein. (**C**) Schematic model of the Lv-CPB protein, with domains, zinc ion-binding site, and catalytic residues labeled as indicated.

**Figure 2 genes-16-00069-f002:**
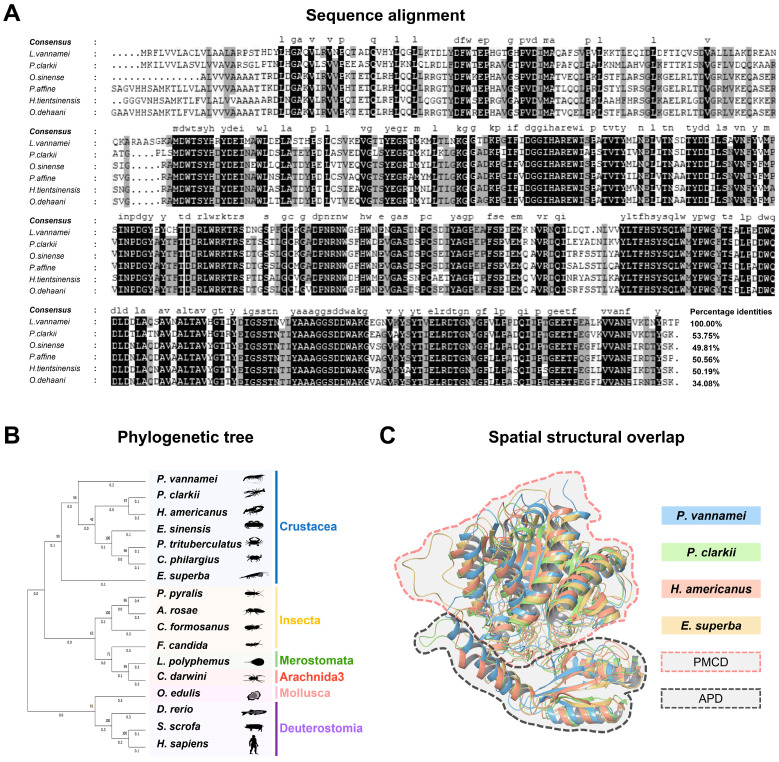
Amino acid sequence alignment, phylogenetic, and 3D structure analysis of Lv-CPB. (**A**) Multiple sequence alignment of Lv-CPB with CPB from other species, where conserved amino acid residues are highlighted in darker shades, with the intensity indicating the degree of conservation. (**B**) Phylogenetic analysis of CPB across various species was conducted using the neighbor-joining method with a bootstrap value of 1000. (**C**) Structural superimposition of CPB protein models from closely related species, with sequences from different species represented in distinct colors. The PMCD is highlighted in red, and the APD is shown in black.

**Figure 3 genes-16-00069-f003:**
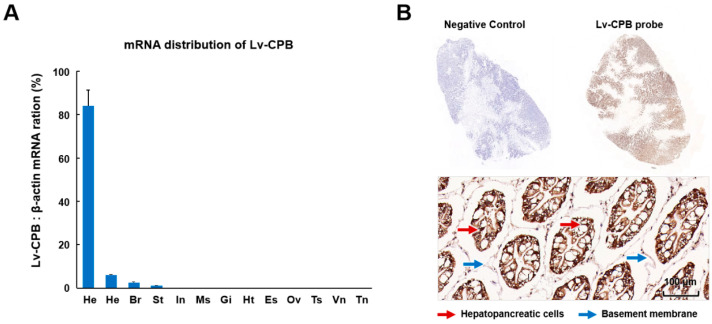
Tissue distribution and localization of Lv-CPB mRNA. (**A**) Expression profile of Lv-CPB mRNA across various tissues of shrimp, including TN (thoracic nerve), VN (ventral nerve cord), Ts (testis), Ov (ovary), Es (esophagus), Ht (heart), Gi (gill), Ms (muscle), In (intestine), St (stomach), Br (brain), He (hemolymph), and Hp (hepatopancreas). Data are presented as mean ± SE from three biological replicates. (**B**) Localization of Lv-CPB mRNA-positive cells in the hepatopancreas of *L. vannamei*. Hematoxylin and eosin (H&E) staining was performed on tissue sections. The negative control for ISH was conducted without a DIG-labeled DNA template. Lv-CPB mRNA-positive cells are distributed throughout the hepatopancreas.

**Figure 4 genes-16-00069-f004:**
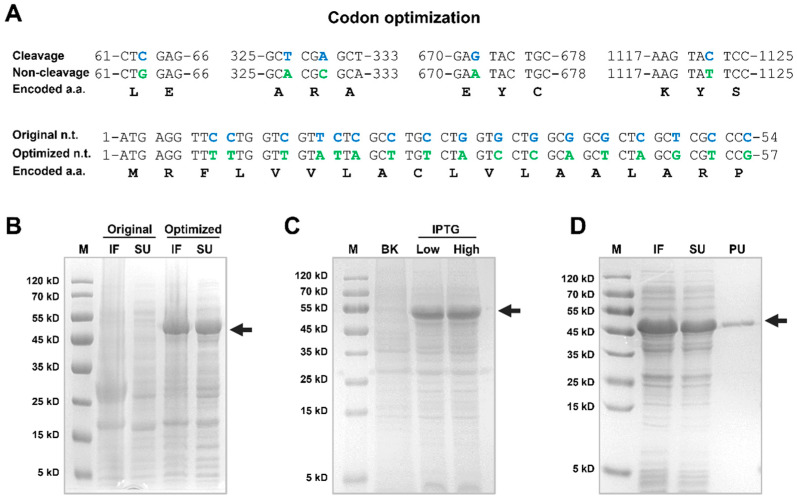
Recombinant expression of the Lv-CPB protein. (**A**) Codon optimization, with original bases highlighted in blue and optimized bases in green. (**B**) Comparison of Lv-CPB protein expression levels in *E. coli* before and after codon optimization, with protein quantified in the soluble supernatant (SU) and inclusion fraction (IF). Arrows indicate target proteins (M: marker). (**C**) Comparison of expression efficiency under different concentrations of IPTG induction. “BK” represents the non-induced control group, “Low” corresponds to induction with 0.2 mM IPTG, and “High” corresponds to induction with 1.0 mM IPTG. (**D**) Comparison of Lv-CPB protein before and after purification, with PU representing the purified sample.

**Figure 5 genes-16-00069-f005:**
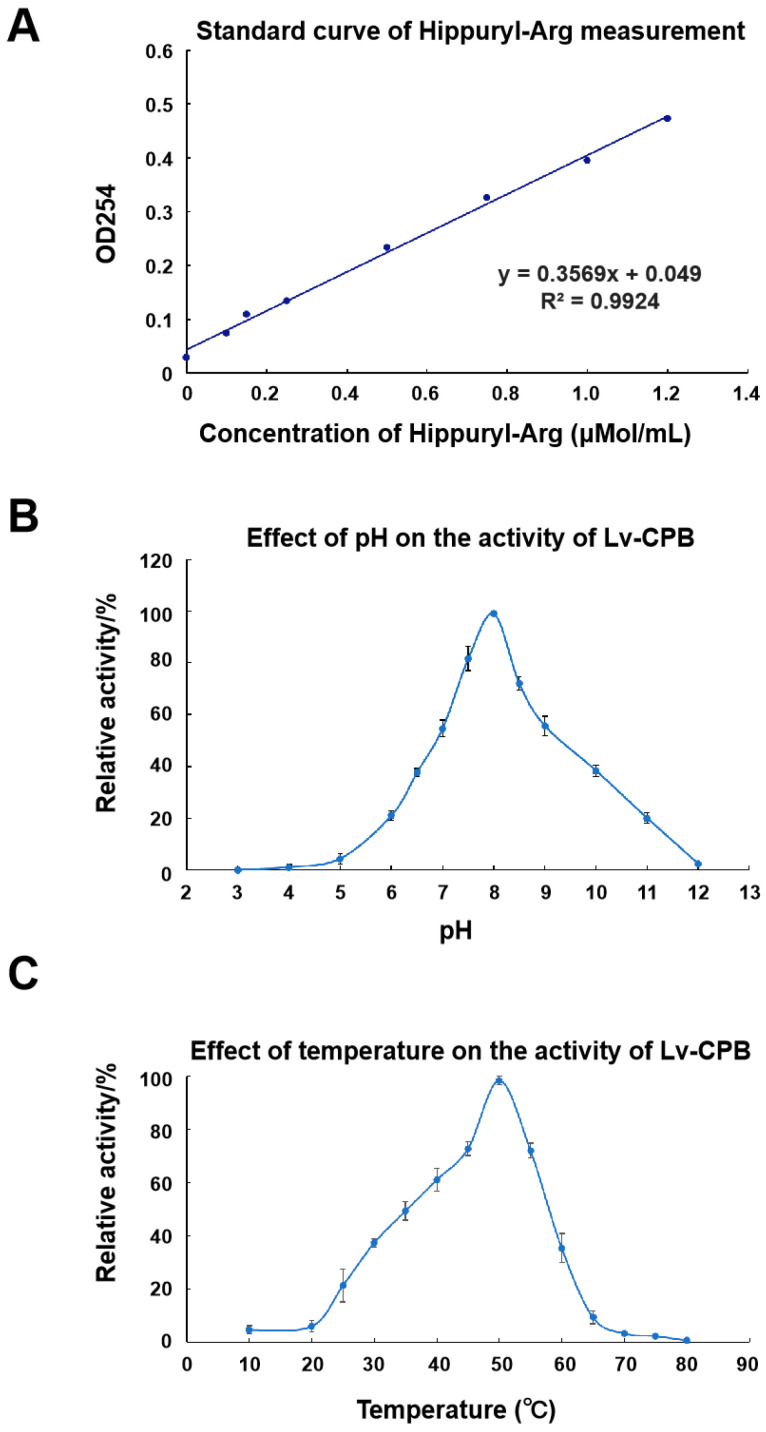
Enzymatic properties of Lv-CPB. (**A**) Standard curve of Lv-CPB’s enzymatic activity, where OD254 indicates the concentration of the catalytic product. (**B**) Activity plot of Lv-CPB as a function of pH. (**C**) Activity plot of Lv-CPB as a function of temperature.

## Data Availability

The data used in this study are publicly available. All relevant data are within the paper and its Appendix A.

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
