# Peer review of "Molecular Characterization, Recombinant Expression, and Functional Analysis of Carboxypeptidase B in Litopenaeus vannamei"

_genes, 2025, doi:10.3390/genes16010069_

Round 1
Reviewer 1 Report
Comments and Suggestions for Authors
The study describes a complete and comprehensive characterization of the Carboxypeptidase B gene in shrimps. The introduction and the discussion are nicely written. I have some important recommendations that I believe should be taken into consideration by the authors.
The authors present correctly in the Introduction the role of Carboxypeptidase B gene in mammals, however since it has been also characterized in Procambarus clarkii, this has to be presented as well.
In the discussion, a comparison of the role in mammals vs the role in shrimp should be provided, maybe in a Table.
What parts of the gene are firstly characterized in invertebrates? Phylogeny, expression, sequence? Or in decapods? E.g. the Please mention
A major addition would be to amplify the gene with the designed primers at DNA level to analyze the whole structure. Are there introns? This is an important lack of the study as it is
Finally, there is a lack of taxonomic nomination consistency in some parts of the manuscript. In the whole manuscript it is referred as Litopenaeus vannamei whereas in Figure 2B and 2C it is referred as P. Vannamei. Please correct
Author Response
Comments 1: The authors present correctly in the Introduction the role of Carboxypeptidase B gene in mammals, however since it has been also characterized in Procambarus clarkii, this has to be presented as well.
Response 1: We sincerely appreciate the reviewer's valuable suggestion. In response, we have incorporated a description of the relevant research on Procambarus clarkii in the Introduction. Currently, the research on P. clarkii is limited to the release of its sequence, and no research team has conducted a comprehensive characterization. In Figure 2C, we performed a comparative analysis of the spatial structures of CPB proteins in P. clarkii, H. americanus, etc. to assess the conservation of CPB protein structures. It should be noted that this analysis was based on software and does not imply that CPB in P. clarkii has been fully characterized. We hope this clarification addresses the reviewer's concern and provides a more accurate context for our study.
Comments 2: In the discussion, a comparison of the role in mammals vs the role in shrimp should be provided, maybe in a Table.
Response 2: In response to your suggestion about providing a comparison of the role of CPB in mammals and shrimp in the discussion section, we have implemented it carefully. In the main text, we have elaborated on the similarities (such as both being involved in hydrolyzing basic amino acids from the C-terminus of peptides for protein digestion) and differences (for example, the role of CPB in the pancreas of mammals has been well studied, while its role in shrimp, especially in Litopenaeus vannamei, is less understood but equally crucial and occurs in the hepatopancreas) between the two. Additionally, we have included sTable 3 in the attachment, which summarizes these similarities and differences. Through such an arrangement, we hope to better respond to your requirement for comparing the two roles and provide readers with more comprehensive and intuitive information.
Comments 3: What parts of the gene are firstly characterized in invertebrates? Phylogeny, expression, sequence? Or in decapods? E.g. the Please mention
Response 3: Thank you for your perceptive question. We have now incorporated the research on the starfish (Asterias amurensis) in the discussion section of our manuscript. This study represents the earliest comprehensive exploration of CPB in marine invertebrates. The research on the starfish CPB involved purifying the enzyme from its pyloric ceca, determining its molecular weight to be approximately 34,000 through polyacrylamide gel electrophoresis, and identifying the optimum pH and temperature for its hydrolysis of a specific substrate. By including this information, we are able to provide a historical context and comparison for our current study on CPB in Litopenaeus vannamei.
This addition not only enriches the content of our paper but also allows for a more in-depth discussion of the diversity and evolution of CPB in different marine invertebrate species. We hope this meets your expectations and enhances the overall value of our research.
Comments 4: A major addition would be to amplify the gene with the designed primers at DNA level to analyze the whole structure. Are there introns? This is an important lack of the study as it is
Response 4: We sincerely thank you for your valuable comments and suggestions. As per your recommendation, we extracted genomic DNA (gDNA) and designed primers based on the exon sequences to amplify and sequence the full-length gene. Through this process, we determined that the Lv-CPB gene of Litopenaeus vannamei spans 12,282 base pairs in the genomic DNA, consisting of 9 exons and 8 introns. The detailed information about the exon-intron boundaries and sizes has been listed in Supplementary Table 3. This content has also been added to the discussion section of the manuscript. Thank you again for your insightful review, which has greatly contributed to the improvement of our work.
Comments 5: Finally, there is a lack of taxonomic nomination consistency in some parts of the manuscript. In the whole manuscript it is referred as Litopenaeus vannamei whereas in Figure 2B and 2C it is referred as P. Vannamei. Please correct
Response 5: Thank you for carefully examining our manuscript. As you recommended, we have systematically updated all relevant content in the manuscript. The "P. vannamei" in Figures 2B and 2C has also been revised to "L. vannamei." We deeply apologize for any confusion caused by this and are grateful for your keen detection of this error. This correction will enhance the clarity and accuracy of our manuscript. Thank you again for assisting us in improving the quality of our work.
Reviewer 2 Report
Comments and Suggestions for Authors
The article entitled Molecular Characterization, Recombinant Expression, and Functional Analysis of Carboxypeptidase B in Litopenaeus vannamei, is interesting and well structurated. However, some minor corrections need to be addressed. See pdf file for more details.
For qPCR primers (STable 1) include addittional columns for amplicon length, Tm, and % of efficiency.
In the case of Supplementary Tables, should be STable instead sTable?
The same for SData 1 (instead sData1?) Please check and be consistent.
Include a ms section with the main abbreviations.

Author Response
Comments 1: For qPCR primers (STable 1) include addittional columns for amplicon length, Tm, and % of efficiency.
Response 1: We are truly grateful for your insightful comments and suggestions. In response to your request, we have added columns for amplicon length, Tm, and % of efficiency in sTable 1 as you recommended. This additional information will provide a more comprehensive and detailed description of the qPCR primers, which we believe will enhance the transparency and reproducibility of our study. Thank you again for your valuable guidance in improving our work.
Comments 2: In the case of Supplementary Tables, should be STable instead sTable?
Response 2: We would like to express our sincere gratitude for your meticulous attention to detail and valuable suggestion. We have fully implemented your advice and have systematically corrected the naming convention of the supplementary tables throughout the manuscript. All instances previously referred to as "sTable" have been updated to "STable" to ensure consistency and adherence to the standard format.
Comments 3: The same for SData 1 (instead sData1?) Please check and be consistent.
Response 3: We have replaced "sData1" with "SData 1" throughout the manuscript to ensure consistency in accordance with your recommendation. Thank you for your valuable guidance.
Comments 4: Include a ms section with the main abbreviations.
Response 4: Thank you for your valuable suggestion. We have added a section in the manuscript dedicated to the main abbreviations as you recommended. In this new section, we have listed all the significant abbreviations used in our study along with their corresponding full terms. We hope this addition meets your expectations and contributes to the overall quality of the paper.
Round 2
Reviewer 1 Report
Comments and Suggestions for Authors
The manuscript is essentially improved, I recommend publication